# WHY DEEP NEURAL NETWORKS FOR FUNCTION APPROXIMATION?

**Shiyu Liang & R. Srikant**
Coordinated Science Laboratory
and
Department of Electrical and Computer Engineering
University of Illinois at Urbana-Champaign
Urbana, IL 61801, USA
`{sliang26,rsrikant}@illinois.edu`

## ABSTRACT

Recently there has been much interest in understanding why deep neural networks are preferred to shallow networks. We show that, for a large class of piecewise smooth functions, the number of neurons needed by a shallow network to approximate a function is exponentially larger than the corresponding number of neurons needed by a deep network for a given degree of function approximation. First, we consider univariate functions on a bounded interval and require a neural network to achieve an approximation error of $\varepsilon$ uniformly over the interval. We show that shallow networks (i.e., networks whose depth does not depend on $\varepsilon$) require $\Omega(\text{poly}(1/\varepsilon))$ neurons while deep networks (i.e., networks whose depth grows with $1/\varepsilon$) require $\mathcal{O}(\text{polylog}(1/\varepsilon))$ neurons. We then extend these results to certain classes of important multivariate functions. Our results are derived for neural networks which use a combination of rectifier linear units (ReLUs) and binary step units, two of the most popular type of activation functions. Our analysis builds on a simple observation: the multiplication of two bits can be represented by a ReLU.

## 1 INTRODUCTION

Neural networks have drawn significant interest from the machine learning community, especially due to their recent empirical successes (see the surveys (Bengio, 2009)). Neural networks are used to build state-of-art systems in various applications such as image recognition, speech recognition, natural language process and others (see, Krizhevsky et al. 2012; Goodfellow et al. 2013; Wan et al. 2013, for example). The result that neural networks are universal approximators is one of the theoretical results most frequently cited to justify the use of neural networks in these applications. Numerous results have shown the universal approximation property of neural networks in approximations of different function classes, (see, e.g., Cybenko 1989; Hornik et al. 1989; Funahashi 1989; Hornik 1991; Chui & Li 1992; Barron 1993; Poggio et al. 2015).

All these results and many others provide upper bounds on the network size and assert that small approximation error can be achieved if the network size is sufficiently large. More recently, there has been much interest in understanding the approximation capabilities of deep versus shallow networks. Delalleau & Bengio (2011) have shown that there exist deep sum-product networks which cannot be approximated by shallow sum-product networks unless they use an exponentially larger amount of units or neurons. Montufar et al. (2014) have shown that the number of linear region increases exponentially with the number of layers in the neural network. Telgarsky (2016) has established such a result for neural networks, which is the subject of this paper. Eldan & Shamir (2015) have shown that, to approximate a specific function, a two-layer network requires an exponential number of neurons in the input dimension, while a three-layer network requires a polynomial number of neurons. These recent papers demonstrate the power of deep networks by showing that depth can lead to an exponential reduction in the number of neurons required, for specific functions or specific neural networks. Our goal here is different: we are interested in function approximation specifically

and would like to show that for a given upper bound on the approximation error, shallow networks require exponentially more neurons than deep networks for a large class of functions.

The multilayer neural networks considered in this paper are allowed to use either rectifier linear units (ReLU) or binary step units (BSU), or any combination of the two. The main contributions of this paper are

- We have shown that, for $\varepsilon$-approximation of functions with enough piecewise smoothness, a multilayer neural network which uses $\Theta(\log(1/\varepsilon))$ layers only needs $\mathcal{O}(\text{poly}\log(1/\varepsilon))$ neurons, while $\Omega(\text{poly}(1/\varepsilon))$ neurons are required by neural networks with $o(\log(1/\varepsilon))$ layers. In other words, shallow networks require exponentially more neurons than a deep network to achieve the level of accuracy for function approximation.

- We have shown that for all differentiable and strongly convex functions, multilayer neural networks need $\Omega(\log(1/\varepsilon))$ neurons to achieve an $\varepsilon$-approximation. Thus, our results for deep networks are tight.

The outline of this paper is as follows. In Section 2, we present necessary definitions and the problem statement. In Section 3, we present upper bounds on network size, while the lower bound is provided in Section 4. Conclusions are presented in Section 5. Around the same time that our paper was uploaded in arxiv, a similar paper was also uploaded in arXiv by Yarotsky (2016). The results in the two papers are similar in spirit, but the details and the general approach are substantially different.

## 2 PRELIMINARIES AND PROBLEM STATEMENT

In this section, we present definitions on feedforward neural networks and formally present the problem statement.

### 2.1 FEEDFORWARD NEURAL NETWORKS

A *feedforward neural network* is composed of layers of computational units and defines a unique function $\tilde{f} : \mathbb{R}^d \to \mathbb{R}$. Let $L$ denote the number of hidden layers, $N_l$ denote the number of units of layer $l$, $N = \sum_{l=1}^{L} N_l$ denote the size of the neural network, vector $\boldsymbol{x} = (x^{(1)}, ..., x^{(d)})$ denote the input of neural network, $z_j^l$ denote the output of the $j$th unit in layer $l$, $w_{i,j}^l$ denote the weight of the edge connecting unit $i$ in layer $l$ and unit $j$ in layer $l+1$, $b_j^l$ denote the bias of the unit $j$ in layer $l$. Then outputs between layers of the feedforward neural network can be characterized by following iterations:

$$z_j^{l+1} = \sigma \left( \sum_{i=1}^{N_l} w_{i,j}^l z_i^l + b_j^{l+1} \right), \quad l \in [L-1], j \in [N_{l+1}],$$

with

$$\text{input layer: } z_j^1 = \sigma \left( \sum_{i=1}^{d} w_{i,j}^0 x^{(i)} + b_j^1 \right), \quad j \in [N_1],$$

$$\text{output layer: } \tilde{f}(\boldsymbol{x}) = \sigma \left( \sum_{i=1}^{N_L} w_{i,j}^L z_i^L + b_j^{L+1} \right).$$

Here, $\sigma(\cdot)$ denotes the activation function and $[n]$ denotes the index set $[n] = \{1, ..., n\}$. In this paper, we only consider two important types of activation functions:

- Rectifier linear unit: $\sigma(x) = \max\{0, x\}, x \in \mathbb{R}$.
- Binary step unit: $\sigma(x) = \mathbb{I}\{x \geq 0\}, x \in \mathbb{R}$.

We call the number of layers and the number of neurons in the network as the *depth* and the *size* of the feedforward neural network, respectively. We use the set $\mathcal{F}(N, L)$ to denote the function set containing all feedforward neural networks of depth $L$, size $N$ and composed of a combination

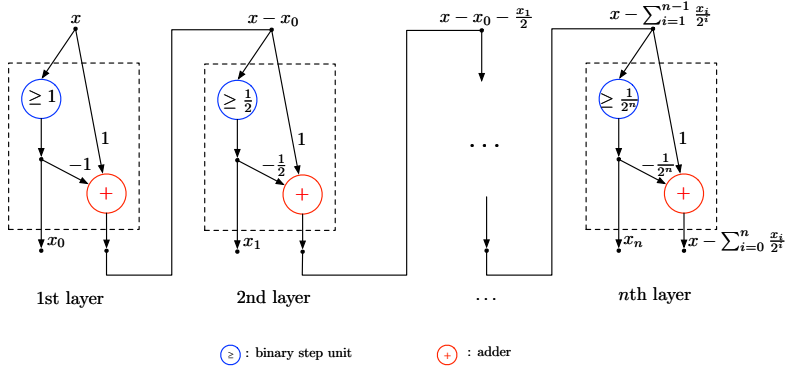

Figure 1: An $n$-layer neural network structure for finding the binary expansion of a number in $[0, 1]$.

of rectifier linear units (ReLUs) and binary step units. We say one feedforward neural network is *deeper* than the other network if and only if it has a larger depth. Through this paper, the terms *feedforward neural network* and *multilayer neural network* are used interchangeably.

## 2.2 PROBLEM STATEMENT

In this paper, we focus on bounds on the size of the feedforward neural network function approximation. Given a function $f$, our goal is to understand whether a multilayer neural network $\tilde{f}$ of depth $L$ and size $N$ exists such that it solves

$$\min_{\tilde{f} \in \mathcal{F}(N,L)} \|f - \tilde{f}\| \le \varepsilon. \tag{1}$$

Specifically, we aim to answer the following questions:

1 Does there exists $L(\varepsilon)$ and $N(\varepsilon)$ such that (1) is satisfied? We will refer to such $L(\varepsilon)$ and $N(\varepsilon)$ as upper bounds on the depth and size of the required neural network.

2 Given a fixed depth $L$, what is the minimum value of $N$ such that (1) is satisfied? We will refer to such an $N$ as a lower bound on the size of a neural network of a given depth $L$.

The first question asks what depth and size are sufficient to guarantee an $\varepsilon$-approximation. The second question asks, for a fixed depth, what is the minimum size of a neural network required to guarantee an $\varepsilon$-approximation. Obviously, tight bounds in the answers to these two questions provide tight bounds on the network size and depth required for function approximation. Besides, solutions to these two questions together can be further used to answer the following question. If a deeper neural network of size $N_d$ and a shallower neural network of size $N_s$ are used to approximate the same function with the same error, then how fast does the ratio $N_d/N_s$ decay to zero as the error decays to zero?

## 3 UPPER BOUNDS ON FUNCTION APPROXIMATIONS

In this section, we present upper bounds on the size of the multilayer neural network which are sufficient for function approximation. Before stating the results, some notations and terminology deserve further explanation. First, the upper bound on the network size represents the number of neurons required at most for approximating a given function with a certain error. Secondly, the notion of the approximation is the $L_\infty$ distance: for two functions $f$ and $g$, the $L_\infty$ distance between these two function is the maximum point-wise disagreement over the cube $[0, 1]^d$.

### 3.1 APPROXIMATION OF UNIVARIATE FUNCTIONS

In this subsection, we present all results on approximating univariate functions. We first present a theorem on the size of the network for approximating a simple quadratic function. As part of the proof, we present the structure of the multilayer feedforward neural network used and show how the neural network parameters are chosen. Results on approximating general functions can be found in Theorem 2 and 4.

**Theorem 1.** *For function $f(x) = x^2, x \in [0,1]$, there exists a multilayer neural network $\tilde{f}(x)$ with $\mathcal{O}\left(\log \frac{1}{\varepsilon}\right)$ layers, $\mathcal{O}\left(\log \frac{1}{\varepsilon}\right)$ binary step units and $\mathcal{O}\left(\log \frac{1}{\varepsilon}\right)$ rectifier linear units such that $|f(x) - \tilde{f}(x)| \le \varepsilon, \quad \forall x \in [0,1].$*

*Proof.* The proof is composed of three parts. For any $x \in [0,1]$, we first use the multilayer neural network to approximate $x$ by its finite binary expansion $\sum_{i=0}^{n} \frac{x_i}{2^i}$. We then construct a 2-layer neural network to implement function $f\left(\sum_{i=0}^{n} \frac{x_i}{2^i}\right)$.

For each $x \in [0,1]$, $x$ can be denoted by its binary expansion $x = \sum_{i=0}^{\infty} \frac{x_i}{2^i}$, where $x_i \in \{0,1\}$ for all $i \ge 0$. It is straightforward to see that the $n$-layer neural network shown in Figure 1 can be used to find $x_0, ..., x_n$.

Next, we implement the function $\tilde{f}(x) = f\left(\sum_{i=0}^{n} \frac{x_i}{2^i}\right)$ by a two-layer neural network. Since $f(x) = x^2$, we then rewrite $\tilde{f}(x)$ as follows:

$$\tilde{f}(x) = \left(\sum_{i=0}^{n} \frac{x_i}{2^i}\right)^2 = \sum_{i=0}^{n} \left[x_i \cdot \left(\frac{1}{2^i} \sum_{j=0}^{n} \frac{x_j}{2^j}\right)\right] = \sum_{i=0}^{n} \max\left(0, 2(x_i - 1) + \frac{1}{2^i} \sum_{j=0}^{n} \frac{x_j}{2^j}\right).$$

The third equality follows from the fact that $x_i \in \{0,1\}$ for all $i$. Therefore, the function $\tilde{f}(x)$ can be implemented by a multilayer network containing a deep structure shown in Figure 1 and another hidden layer with $n$ rectifier linear units. This multilayer neural network has $\mathcal{O}(n)$ layers, $\mathcal{O}(n)$ binary step units and $\mathcal{O}(n)$ rectifier linear units.

Finally, we consider the approximation error of this multilayer neural network,

$$|f(x) - \tilde{f}(x)| = \left|x^2 - \left(\sum_{i=0}^{n} \frac{x_i}{2^i}\right)^2\right| \le 2\left|x - \sum_{i=0}^{n} \frac{x_i}{2^i}\right| = 2\left|\sum_{i=n+1}^{\infty} \frac{x_i}{2^i}\right| \le \frac{1}{2^{n-1}}.$$

Therefore, in order to achieve $\varepsilon$-approximation error, one should choose $n = \left\lceil \log_2 \frac{1}{\varepsilon} \right\rceil + 1$. In summary, the deep neural network has $\mathcal{O}\left(\log \frac{1}{\varepsilon}\right)$ layers, $\mathcal{O}\left(\log \frac{1}{\varepsilon}\right)$ binary step units and $\mathcal{O}\left(\log\left(\frac{1}{\varepsilon}\right)\right)$ rectifier linear units. $\square$

Next, a theorem on the size of the network for approximating general polynomials is given as follows.

**Theorem 2.** *For polynomials $f(x) = \sum_{i=0}^{p} a_i x^i$, $x \in [0,1]$ and $\sum_{i=1}^{p} |a_i| \le 1$, there exists a multilayer neural network $\tilde{f}(x)$ with $\mathcal{O}\left(p + \log \frac{p}{\varepsilon}\right)$ layers, $\mathcal{O}\left(\log \frac{p}{\varepsilon}\right)$ binary step units and $\mathcal{O}\left(p \log \frac{p}{\varepsilon}\right)$ rectifier linear units such that $|f(x) - \tilde{f}(x)| \le \varepsilon, \forall x \in [0,1].$*

*Proof.* The proof is composed of three parts. We first use the deep structure shown in Figure 1 to find the $n$-bit binary expansion $\sum_{i=0}^{n} a_i x^i$ of $x$. Then we construct a multilayer network to approximate polynomials $g_i(x) = x^i$, $i = 1, ..., p$. Finally, we analyze the approximation error.

Using the same deep structure shown in Figure 1, we could find the binary expansion sequence $\{x_0, ..., x_n\}$. In this step, we used $n$ binary steps units in total. Now we rewrite $g_{m+1}(\sum_{i=0}^{n} \frac{x_i}{2^n})$,

$$g_{m+1}\left(\sum_{i=0}^{n} \frac{x_i}{2^i}\right) = \sum_{j=0}^{n} \left[x_j \cdot \frac{1}{2^j} g_m\left(\sum_{i=0}^{n} \frac{x_i}{2^i}\right)\right] = \sum_{j=0}^{n} \max\left[2(x_j - 1) + \frac{1}{2^j} g_m\left(\sum_{i=0}^{n} \frac{x_i}{2^i}\right), 0\right].$$

(2)

Clearly, the equation (2) defines iterations between the outputs of neighbor layers. Therefore, the deep neural network shown in Figure 2 can be used to implement the iteration given by (2). Further, to implement this network, one should use $\mathcal{O}(p)$ layers with $\mathcal{O}(pn)$ rectifier linear units in total. We now define the output of the multilayer neural network as $\tilde{f}(x) = \sum_{i=0}^{p} a_i g_i\left(\sum_{j=0}^{n} \frac{x_j}{2^j}\right)$. For this multilayer network, the approximation error is

$$|f(x) - \tilde{f}(x)| = \left|\sum_{i=0}^{p} a_i g_i\left(\sum_{j=0}^{n} \frac{x_j}{2^j}\right) - \sum_{i=0}^{p} a_i x^i\right| \le \sum_{i=0}^{p} \left[|a_i| \cdot \left|g_i\left(\sum_{j=0}^{n} \frac{x_j}{2^j}\right) - x^i\right|\right] \le \frac{p}{2^{n-1}}$$

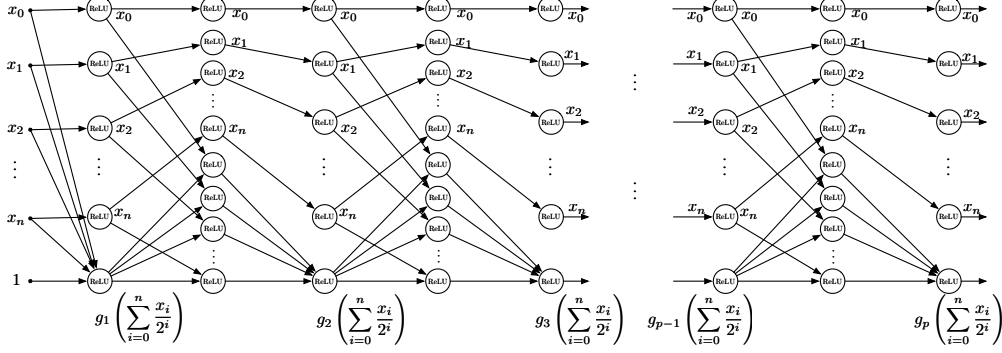

Figure 2: The implementation of polynomial function

This indicates, to achieve $\varepsilon$-approximation error, one should choose $n = \left\lceil \log \frac{p}{\varepsilon} \right\rceil + 1$. Besides, since we used $\mathcal{O}(n + p)$ layers with $\mathcal{O}(n)$ binary step units and $\mathcal{O}(pn)$ rectifier linear units in total, this multilayer neural network thus has $\mathcal{O}\left(p + \log \frac{p}{\varepsilon}\right)$ layers, $\mathcal{O}\left(\log \frac{p}{\varepsilon}\right)$ binary step units and $\mathcal{O}\left(p \log \frac{p}{\varepsilon}\right)$ rectifier linear units. $\qquad \square$

In Theorem 2, we have shown an upper bound on the size of multilayer neural network for approximating polynomials. We can easily observe that the number of neurons in network grows as $p \log p$ with respect to $p$, the degree of the polynomial. We note that both Andoni et al. (2014) and Barron (1993) showed the sizes of the networks grow exponentially with respect to $p$ if only 3-layer neural networks are allowed to be used in approximating polynomials.

Besides, every function $f$ with $p + 1$ continuous derivatives on a bounded set can be approximated easily with a polynomial with degree $p$. This is shown by the following well known result of Lagrangian interpolation. By this result, we could further generalize Theorem 2. The proof can be found in the reference (Gil et al., 2007).

**Lemma 3** (**Lagrangian interpolation at Chebyshev points**). *If a function $f$ is defined at points $z_0, ..., z_n$, $z_i = \cos((i + 1/2)\pi/(n + 1))$, $i \in [n]$, there exists a polynomial of degree not more than $n$ such that $P_n(z_i) = f(z_i)$, $i = 0, ..., n$. This polynomial is given by $P_n(x) = \sum_{i=0}^{n} f(z_i) L_i(x)$ where $L_i(x) = \frac{\pi_{n+1}(x)}{(x - z_i)\pi'_{n+1}(z_i)}$ and $\pi_{n+1}(x) = \prod_{j=0}^{n}(x - z_j)$. Additionally, if $f$ is continuous on $[-1, 1]$ and $n + 1$ times differentiable in $(-1, 1)$, then*

$$\|R_n\| = \|f - P_n\| \leq \frac{1}{2^n (n + 1)!} \left\| f^{(n+1)} \right\|,$$

*where $f^{(n)}(x)$ is the derivative of $f$ of the $n$th order and the norm $\|f\|$ is the $l_\infty$ norm $\|f\| = \max_{x \in [-1,1]} f(x)$.*

Then the upper bound on the network size for approximating more general functions follows directly from Theorem 2 and Lemma 3.

**Theorem 4.** *Assume that function $f$ is continuous on $[0, 1]$ and $\left\lceil \log \frac{2}{\varepsilon} \right\rceil + 1$ times differentiable in $(0, 1)$. Let $f^{(n)}$ denote the derivative of $f$ of $n$th order and $\|f\| = \max_{x \in [0,1]} f(x)$. If $\left\| f^{(n)} \right\| \leq n!$ holds for all $n \in \left[\left\lceil \log \frac{2}{\varepsilon} \right\rceil + 1\right]$, then there exists a deep neural network $\tilde{f}$ with $\mathcal{O}\left(\log \frac{1}{\varepsilon}\right)$ layers, $\mathcal{O}\left(\log \frac{1}{\varepsilon}\right)$ binary step units, $\mathcal{O}\left(\left(\log \frac{1}{\varepsilon}\right)^2\right)$ rectifier linear units such that $\left\| f - \tilde{f} \right\| \leq \varepsilon$.*

*Proof.* Let $N = \left\lceil \log \frac{2}{\varepsilon} \right\rceil$. From Lemma 3, it follows that there exists polynomial $P_N$ of degree $N$ such that for any $x \in [0, 1]$,

$$|f(x) - P_N(x)| \leq \frac{\left\| f^{(N+1)} \right\|}{2^N (N + 1)!} \leq \frac{1}{2^N}.$$

Let $x_0, ..., x_N$ denote the first $N + 1$ bits of the binary expansion of $x$ and define $\tilde{f}(x) = P_N\left(\sum_{i=0}^{N} \frac{x_i}{2^N}\right)$. In the following, we first analyze the approximation error of $\tilde{f}$ and next

show the implementation of this function. Let $\tilde{x} = \sum_{i=0}^{N} \frac{x_i}{2^i}$. The error can now be upper bounded by

$$|f(x) - \tilde{f}(x)| = |f(x) - P_N(\tilde{x})| \leq |f(x) - f(\tilde{x})| + |f(\tilde{x}) - P_N(\tilde{x})|$$

$$\leq \left\| f^{(1)} \right\| \cdot \left| x - \sum_{i=0}^{N} \frac{x_i}{2^i} \right| + \frac{1}{2^N} \leq \frac{1}{2^N} + \frac{1}{2^N} \leq \varepsilon$$

In the following, we describe the implementation of $\tilde{f}$ by a multilayer neural network. Since $P_N$ is a polynomial of degree $N$, function $\tilde{f}$ can be rewritten as

$$\tilde{f}(x) = P_N \left( \sum_{i=0}^{N} \frac{x_i}{2^i} \right) = \sum_{n=0}^{N} c_n g_n \left( \sum_{i=0}^{N} \frac{x_i}{2^i} \right)$$

for some coefficients $c_0, ..., c_N$ and $g_n = x^n$, $n \in [N]$. Hence, the multilayer neural network shown in the Figure 2 can be used to implement $\tilde{f}(x)$. Notice that the network uses $\mathcal{O}(N)$ layers with $\mathcal{O}(N)$ binary step units in total to decode $x_0,...,x_N$ and $\mathcal{O}(N)$ layers with $\mathcal{O}(N^2)$ rectifier linear units in total to construct the polynomial $P_N$. Substituting $N = \left\lceil \log \frac{2}{\varepsilon} \right\rceil$, we have proved the theorem. $\square$

**Remark:** Note that, to implement the architecture in Figure 2 using the definition of a feedforward neural network in Section 2, we need the $g_i$, $i \in [p]$ at the output. This can be accomplished by using $\mathcal{O}(p^2)$ additional ReLUs. Since $p = \mathcal{O}(\log(1/\varepsilon))$, this doesn't change the order result in Theorem 4.

Theorem 4 shows that any function $f$ with enough smoothness can be approximated by a multilayer neural network containing $\text{polylog}\left(\frac{1}{\varepsilon}\right)$ neurons with $\varepsilon$ error. Further, Theorem 4 can be used to show that for functions $h_1,...,h_k$ with enough smoothness, then linear combinations, multiplications and compositions of these functions can as well be approximated by multilayer neural networks containing $\text{polylog}\left(\frac{1}{\varepsilon}\right)$ neurons with $\varepsilon$ error. Specific results are given in the following corollaries.

**Corollary 5** (Function addition)**.** *Suppose that all functions $h_1, ..., h_k$ satisfy the conditions in Theorem 4, and the vector $\boldsymbol{\beta} \in \{\boldsymbol{\omega} \in \mathbb{R}^k : \|\boldsymbol{\omega}\|_1 = 1\}$, then for the linear combination $f = \sum_{i=1}^{k} \beta_i h_i$, there exists a deep neural network $\tilde{f}$ with $\mathcal{O}\left(\log \frac{1}{\varepsilon}\right)$ layers, $\mathcal{O}\left(\log \frac{1}{\varepsilon}\right)$ binary step units, $\mathcal{O}\left(\left(\log \frac{1}{\varepsilon}\right)^2\right)$ rectifier linear units such that $|f(x) - \tilde{f}| \leq \varepsilon$, $\forall x \in [0,1]$.*

**Remark:** Clearly, Corollary 5 follows directly from the fact that the linear combination $f$ satisfies the conditions in Theorem 4 if all the functions $h_1,...,h_k$ satisfy those conditions. We note here that the upper bound on the network size for approximating linear combinations is independent of $k$, the number of component functions.

**Corollary 6** (Function multiplication)**.** *Suppose that all functions $h_1,...,h_k$ are continuous on $[0,1]$ and $\left\lceil 4k \log_2 4k + 4k + 2 \log_2 \frac{2}{\varepsilon} \right\rceil + 1$ times differentiable in $(0,1)$. If $\|h_i^{(n)}\| \leq n!$ holds for all $i \in [k]$ and $n \in \left[ \left\lceil 4k \log_2 4k + 4k + 2 \log_2 \frac{2}{\varepsilon} \right\rceil + 1 \right]$ then for the multiplication $f = \prod_{i=1}^{k} h_i$, there exists a multilayer neural network $\tilde{f}$ with $\mathcal{O}\left(k \log k + \log \frac{1}{\varepsilon}\right)$ layers, $\mathcal{O}\left(k \log k + \log \frac{1}{\varepsilon}\right)$ binary step units and $\mathcal{O}\left((k \log k)^2 + \left(\log \frac{1}{\varepsilon}\right)^2\right)$ rectifier linear units such that $|f(x) - \tilde{f}(x)| \leq \varepsilon$, $\forall x \in [0,1]$.*

**Corollary 7** (Function composition)**.** *Suppose that all functions $h_1, ..., h_k : [0,1] \to [0,1]$ satisfy the conditions in Theorem 4, then for the composition $f = h_1 \circ h_2 \circ ... \circ h_k$, there exists a multilayer neural network $\tilde{f}$ with $\mathcal{O}\left(k \log k \log \frac{1}{\varepsilon} + \log k \left(\log \frac{1}{\varepsilon}\right)^2\right)$ layers, $\mathcal{O}\left(k \log k \log \frac{1}{\varepsilon} + \log k \left(\log \frac{1}{\varepsilon}\right)^2\right)$ binary step units and $\mathcal{O}\left(k^2 \left(\log \frac{1}{\varepsilon}\right)^2 + \left(\log \frac{1}{\varepsilon}\right)^4\right)$ rectifier linear units such that $|f(x) - \tilde{f}(x)| \leq \varepsilon$, $\forall x \in [0,1]$.*

**Remark:** Proofs of Corollary 6 and 7 can be found in the appendix. We observe that different from the case of linear combinations, the upper bound on the network size grows as $k^2 \log^2 k$ in the case of function multiplications and grows as $k^2 \left(\log \frac{1}{\varepsilon}\right)^2$ in the case of function compositions where $k$ is the number of component functions.

In this subsection, we have shown a $\text{polylog}\left(\frac{1}{\varepsilon}\right)$ upper bound on the network size for $\varepsilon$-approximation of both univariate polynomials and general univariate functions with enough smoothness. Besides, we have shown that linear combinations, multiplications and compositions of univariate functions with enough smoothness can as well be approximated with $\varepsilon$ error by a multilayer neural network of size $\text{polylog}\left(\frac{1}{\varepsilon}\right)$. In the next subsection, we will show the upper bound on the network size for approximating multivariate functions.

## 3.2 Approximation of multivariate functions

In this subsection, we present all results on approximating multivariate functions. We first present a theorem on the upper bound on the neural network size for approximating a product of multivariate linear functions. We next present a theorem on the upper bound on the neural network size for approximating general multivariate polynomial functions. Finally, similar to the results in the univariate case, we present the upper bound on the neural network size for approximating the linear combination, the multiplication and the composition of multivariate functions with enough smoothness.

**Theorem 8.** *Let $W = \{w \in \mathbb{R}^d : \|w\|_1 = 1\}$. For $f(x) = \prod_{i=1}^{p}\left(w_i^T x\right)$, $x \in [0,1]^d$ and $w_i \in W$, $i = 1,...,p$, there exists a deep neural network $\tilde{f}(x)$ with $\mathcal{O}\left(p + \log\frac{pd}{\varepsilon}\right)$ layers and $\mathcal{O}\left(\log\frac{pd}{\varepsilon}\right)$ binary step units and $\mathcal{O}\left(pd\log\frac{pd}{\varepsilon}\right)$ rectifier linear units such that $|f(x) - \tilde{f}(x)| \leq \varepsilon$, $\forall x \in [0,1]^d$.*

Theorem 8 shows an upper bound on the network size for $\varepsilon$-approximation of a product of multivariate linear functions. Furthermore, since any general multivariate polynomial can be viewed as a linear combination of products, the result on general multivariate polynomials directly follows from Theorem 8.

**Theorem 9.** *Let the multi-index vector $\boldsymbol{\alpha} = (\alpha_1,...,\alpha_d)$, the norm $|\boldsymbol{\alpha}| = \alpha_1+...+\alpha_d$, the coefficient $C_{\boldsymbol{\alpha}} = C_{\alpha_1...\alpha_d}$, the input vector $x = (x^{(1)},...,x^{(d)})$ and the multinomial $x^{\boldsymbol{\alpha}} = x^{(1)^{\alpha_1}}...x^{(d)^{\alpha_d}}$. For positive integer $p$ and polynomial $f(x) = \sum_{\boldsymbol{\alpha}:|\boldsymbol{\alpha}|\leq p} C_{\boldsymbol{\alpha}} x^{\boldsymbol{\alpha}}$, $x \in [0,1]^d$ and $\sum_{\boldsymbol{\alpha}:|\boldsymbol{\alpha}|\leq p}|C_{\boldsymbol{\alpha}}| \leq 1$, there exists a deep neural network $\tilde{f}(x)$ of depth $\mathcal{O}\left(p + \log\frac{dp}{\varepsilon}\right)$ and size $N(d,p,\varepsilon)$ such that $|f(x) - f(\tilde{x})| \leq \varepsilon$, where*

$$N(d,p,\varepsilon) = p^2\binom{p+d-1}{d-1}\log\frac{pd}{\varepsilon}.$$

**Remark:** The proof is given in the appendix. By further analyzing the results on the network size, we obtain the following results: (a) fixing degree $p$, $N(d,\varepsilon) = \mathcal{O}\left(d^{p+1}\log\frac{d}{\varepsilon}\right)$ as $d \to \infty$ and (b) fixing input dimension $d$, $N(p,\varepsilon) = \mathcal{O}\left(p^d\log\frac{p}{\varepsilon}\right)$ as $p \to \infty$. Similar results on approximating multivariate polynomials were obtained by Andoni et al. (2014) and Barron (1993). Barron (1993) showed that on can use a 3-layer neural network to approximate any multivariate polynomial with degree $p$, dimension $d$ and network size $d^p/\varepsilon^2$. Andoni et al. (2014) showed that one could use the gradient descent to train a 3-layer neural network of size $d^{2p}/\varepsilon^2$ to approximate any multivariate polynomial. However, Theorem 9 shows that the deep neural network could reduce the network size from $\mathcal{O}(1/\varepsilon)$ to $\mathcal{O}(\log\frac{1}{\varepsilon})$ for the same $\varepsilon$ error. Besides, for a fixed input dimension $d$, the size of the 3-layer neural network used by Andoni et al. (2014) and Barron (1993) grows exponentially with respect to the degree $p$. However, the size of the deep neural network shown in Theorem 9 grows only polynomially with respect to the degree. Therefore, the deep neural network could reduce the network size from $\mathcal{O}(\exp(p))$ to $\mathcal{O}(\text{poly}(p))$ when the degree $p$ becomes large.

Theorem 9 shows an upper bound on the network size for approximating multivariate polynomials. Further, by combining Theorem 4 and Corollary 7, we could obtain an upper bound on the network size for approximating more general functions. The results are shown in the following corollary.

**Corollary 10.** *Assume that all univariate functions $h_1,...,h_k : [0,1] \to [0,1]$, $k \geq 1$, satisfy the conditions in Theorem 4. Assume that the multivariate polynomial $l(x) : [0,1]^d \to [0,1]$ is of degree $p$. For composition $f = h_1 \circ h_2 \circ ... \circ h_k \circ l(x)$, there exists a multilayer neural network $\tilde{f}$ of depth $\mathcal{O}\left(p + \log d + k\log k\log\frac{1}{\varepsilon} + \log k\left(\log\frac{1}{\varepsilon}\right)^2\right)$ and of size $N(k,p,d,\varepsilon)$ such that $|\tilde{f}(x)-f(x)| \leq \varepsilon$ for $\forall x \in [0,1]^d$, where*

$$N(k, p, d, \varepsilon) = \mathcal{O}\left(p^2 \binom{p + d - 1}{d - 1} \log \frac{pd}{\varepsilon} + k^2 \left(\log \frac{1}{\varepsilon}\right)^2 + \left(\log \frac{1}{\varepsilon}\right)^4\right).$$

**Remark**: Corollary 10 shows an upper bound on network size for approximating compositions of multivariate polynomials and general univariate functions. The upper bound can be loose due to the assumption that $l(\boldsymbol{x})$ is a general multivariate polynomials of degree $p$. For some specific cases, the upper bound can be much smaller. We present two specific examples in the Appendix H and I.

In this subsection, we have shown that a similar polylog$\left(\frac{1}{\varepsilon}\right)$ upper bound on the network size for $\varepsilon$-approximation of general multivariate polynomials and functions which are compositions of univariate functions and multivariate polynomials.

The results in this section can be used to find a multilayer neural network of size polylog$\left(\frac{1}{\varepsilon}\right)$ which provides an approximation error of at most $\varepsilon$. In the next section, we will present lower bounds on the network size for approximating both univariate and multivariate functions. The lower bound together with the upper bound shows a tight bound on the network size required for function approximations.

While we have presented results in both the univariate and multivariate cases for smooth functions, the results automatically extend to functions that are piecewise smooth, with a finite number of pieces. In other words, if the domain of the function is partitioned into regions, and the function is sufficiently smooth (in the sense described in the Theorems and Corollaries earlier) in each of the regions, then the results essentially remain unchanged except for an additional factor which will depend on the number of regions in the domain.

## 4 LOWER BOUNDS ON FUNCTION APPROXIMATIONS

In this section, we present lower bounds on the network size in function for certain classes of functions. Next, by combining the lower bounds and the upper bounds shown in the previous section, we could analytically show the advantages of deeper neural networks over shallower ones. The theorem below is inspired by a similar result (DasGupta & Schnitger, 1993) for univariate quadratic functions, where it is stated without a proof. Here we show that the result extends to general multivariate strongly convex functions.

**Theorem 11.** *Assume function $f : [0, 1]^d \to \mathbb{R}$ is differentiable and strongly convex with parameter $\mu$. Assume the multilayer neural network $\tilde{f}$ is composed of rectifier linear units and binary step units. If $|f(\boldsymbol{x}) - \tilde{f}(\boldsymbol{x})| \le \varepsilon, \forall \boldsymbol{x} \in [0, 1]^d$, then the network size $N \ge \log_2\left(\frac{\mu}{16\varepsilon}\right)$.*

**Remark:** The proof is in the Appendix F. Theorem 11 shows that every strongly convex function cannot be approximated with error $\varepsilon$ by any multilayer neural network with rectifier linear units and binary step units and of size smaller than $\log_2(\mu/\varepsilon) - 4$. Theorem 11 together with Theorem 1 directly shows that to approximate quadratic function $f(x) = x^2$ with error $\varepsilon$, the network size should be of order $\Theta\left(\log \frac{1}{\varepsilon}\right)$. Further, by combining Theorem 11 and Theorem 4, we could analytically show the benefits of deeper neural networks. The result is given in the following corollary.

**Corollary 12.** *Assume that univariate function $f$ satisfies conditions in both Theorem 4 and Theorem 11. If a neural network $\tilde{f}_s$ is of depth $L_s = o\left(\log \frac{1}{\varepsilon}\right)$, size $N_s$ and $|f(x) - \tilde{f}_s(x)| \le \varepsilon$, for $\forall x \in [0, 1]$, then there exists a deeper neural network $\tilde{f}_d(x)$ of depth $\Theta\left(\log \frac{1}{\varepsilon}\right)$, size $N_d = \mathcal{O}(L_s^2 \log^2 N_s)$ such that $|f(x) - \tilde{f}_d(x)| \le \varepsilon, \forall x \in [0, 1]$.*

**Remarks:** (i) The strong convexity requirement can be relaxed: the result obviously holds if the function is strongly concave and it also holds if the function consists of pieces which are strongly convex or strongly concave. (ii) Corollary 12 shows that in the approximation of the same function, the size of the deep neural network $N_s$ is only of polynomially logarithmic order of the size of the shallow neural network $N_d$, *i.e.*, $N_d = \mathcal{O}(\text{polylog}(N_s))$. Similar results can be obtained for multivariate functions on the type considered in Section 3.2.

## 5 CONCLUSIONS

In this paper, we have shown that an exponentially large number of neurons are needed for function approximation using shallow networks, when compared to deep networks. The results are established for a large class of smooth univariate and multivariate functions. Our results are established for the case of feedforward neural networks with ReLUs and binary step units.

ACKNOWLEDGMENTS

The research reported here was supported by NSF Grants CIF 14-09106, ECCS 16-09370, and ARO Grant W911NF-16-1-0259.

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

## APPENDIX A   PROOF OF COROLLARY 5

*Proof.* By Theorem 4, for each $h_i$, $i = 1, ..., k$, there exists a multilayer neural network $\tilde{h}_i$ such that $|h_i(x) - \tilde{h}(x)| \leq \varepsilon$ for any $x \in [0, 1]$. Let

$$\tilde{f}(x) = \sum_{i=1}^{k} \beta_i \tilde{h}_i(x).$$

Then the approximation error is upper bounded by

$$|f(x) - \tilde{f}(x)| = \left| \sum_{i=1}^{k} \beta_i h_i(x) \right| \leq \sum_{i=1}^{k} |\beta_i| \cdot |h_i(x) - \tilde{h}(x)| = \varepsilon.$$

Now we compute the size of the multilayer neural network $\tilde{f}$. Let $N = \lceil \log \frac{2}{\varepsilon} \rceil$ and $\sum_{i=0}^{N} \frac{x_i}{2^i}$ be the binary expansion of $x$. Since $\tilde{h}_i(x)$ has a form of

$$\tilde{h}_i(x) = \sum_{j=0}^{N} c_{ij} g_j \left( \sum_{i=0}^{N} \frac{x_i}{2^i} \right),$$

where $g_j(x) = x^j$, then $\tilde{f}$ should has a form of

$$\tilde{f}(x) = \sum_{i=1}^{k} \beta_i \left[ \sum_{j=0}^{N} c_{ij} g_j \left( \sum_{i=0}^{N} \frac{x_i}{2^i} \right) \right]$$

and can be further rewritten as

$$\tilde{f}(x) = \sum_{j=0}^{N} \left[ \left( \sum_{i=1}^{k} c_{ij} \beta_i \right) \cdot g_j \left( \sum_{i=0}^{N} \frac{x_i}{2^i} \right) \right] \triangleq \sum_{j=0}^{N} c'_j g_j \left( \sum_{i=0}^{N} \frac{x_i}{2^i} \right),$$

where $c'_j = \sum_i c_{ij} \beta_i$. Therefore, $\tilde{f}$ can be implemented by a multilayer neural network shown in Figure 2 and this network has at most $\mathcal{O}\left( \log \frac{1}{\varepsilon} \right)$ layers, $\mathcal{O}\left( \log \frac{1}{\varepsilon} \right)$ binary step units, $\mathcal{O}\left( \left( \log \frac{1}{\varepsilon} \right)^2 \right)$ rectifier linear units. $\qquad \square$

## APPENDIX B   PROOF OF COROLLARY 6

*Proof.* Since $f(x) = h_1(x) h_2(x) ... h_k(x)$, then the derivative of $f$ of order $n$ is

$$f^{(n)} = \sum_{\substack{\alpha_1 + ... + \alpha_k = n \\ \alpha_1 \geq 0, ..., \alpha_k \geq 0}} \frac{n!}{\alpha_1! \alpha_2! ... \alpha_k!} h_1^{(\alpha_1)} h_2^{(\alpha_2)} ... h_k^{(\alpha_k)}.$$

By the assumption that $\left\| h_i^{(\alpha_i)} \right\| \leq \alpha_i!$ holds for $i = 1, ..., k$, then we have

$$\left\| f^{(n)} \right\| \leq \sum_{\substack{\alpha_1 + ... + \alpha_k = n \\ \alpha_1 \geq 0, ..., \alpha_k \geq 0}} \frac{n!}{\alpha_1! \alpha_2! ... \alpha_k!} \left\| h_1^{(\alpha_1)} h_2^{(\alpha_2)} ... h_k^{(\alpha_k)} \right\| \leq \binom{n + k - 1}{k - 1} n!.$$

Then from Theorem 4, it follows that there exists a polynomial of $P_N$ degree $N$ that

$$\| R_N \| = \| f - P_N \| \leq \frac{\| f^{(N+1)} \|}{(N+1)! 2^N} \leq \frac{1}{2^N} \binom{N+k}{k-1}.$$

Since

$$\binom{N+k}{k-1} \leq \frac{(N+k)^{N+k}}{(k-1)^{k-1}(N+1)^{N+1}} = \left( \frac{N+k}{k-1} \right)^{k-1} \left( 1 + \frac{k-1}{N+1} \right)^{N+1} \leq \left( \frac{e(N+k)}{k-1} \right)^{k-1}$$

then the error has an upper bound of

$$\|R_N\| \le \frac{(eN)^k}{2^N} \le 2^{2k + k \log_2 N - N}. \tag{3}$$

Since we need to bound

$$\|R_N\| \le \frac{\varepsilon}{2},$$

then we need to choose $N$ such that

$$N \ge k \log_2 N + 2k + \log_2 \frac{2}{\varepsilon}.$$

Thus, $N$ can be chosen such that

$$N \ge 2k \log_2 N \quad \text{and} \quad N \ge 4k + 2 \log_2 \frac{2}{\varepsilon}.$$

Further, function $l(x) = x / \log_2 x$ is monotonically increasing on $[e, \infty)$ and

$$l(4k \log_2 4k) = \frac{4k \log_2 4k}{\log_2 4k + \log_2 \log_2 4k} \ge \frac{4k \log_2 4k}{\log_2 4k + \log_2 4k} = 2k.$$

Therefore, to suffice the inequality (3), one should should choose

$$N \ge 4k \log_2 4k + 4k + 2 \log_2 \frac{2}{\varepsilon}.$$

Since $N = \left\lceil 4k \log_2 4k + 4k + 2 \log_2 \frac{2}{\varepsilon} \right\rceil$ by assumptions, then there exists a polynomial $P_N$ of degree $N$ such that

$$\|f - P_N\| \le \frac{\varepsilon}{2}.$$

Let $\sum_{i=0}^{N} \frac{x_i}{2^i}$ denote the binary expansion of $x$ and let

$$\tilde{f}(x) = P_N \left( \sum_{i=0}^{N} \frac{x_i}{2^i} \right).$$

The approximation error is

$$|\tilde{f}(x) - f(x)| \le \left| f(x) - f\left( \sum_{i=0}^{N} \frac{x_i}{2^i} \right) \right| + \left| f\left( \sum_{i=0}^{N} \frac{x_i}{2^i} \right) - P_N \left( \sum_{i=0}^{N} \frac{x_i}{2^i} \right) \right|$$

$$\le \|f(1)\| \left| x - \sum_{i=0}^{N} \frac{x_i}{2^i} \right| + \frac{\varepsilon}{2} \le \varepsilon$$

Further, function $\tilde{f}$ can be implemented by a multilayer neural network shown in Figure 2 and this network has at most $\mathcal{O}(N)$ layers, $\mathcal{O}(N)$ binary step units and $\mathcal{O}(N^2)$ rectifier linear units. □

## APPENDIX C   PROOF OF COROLLARY 7

*Proof.* We prove this theorem by induction. Define function $F_m = h_1 \circ ... \circ h_m$, $m = 1, ..., k$. Let $T_1(m) \log_3 \frac{3^m}{\varepsilon}$, $T_2(m) \log_3 \frac{3^m}{\varepsilon}$ and $T_3(m) \left( \log_3 \frac{3^m}{\varepsilon} \right)^2$ denote the number of layers, the number of binary step units and the number of rectifier linear units required at most for $\varepsilon$-approximation of $F_m$, respectively. By Theorem 4, for $m = 1$, there exists a multilayer neural network $\tilde{F}_1$ with at most $T_1(1) \log_3 \frac{3}{\varepsilon}$ layers, $T_2(1) \log_3 \frac{3}{\varepsilon}$ binary step units and $T_3(1) \left( \log_3 \frac{3}{\varepsilon} \right)^2$ rectifier linear units such that

$$|F_1(x) - \tilde{F}_1(x)| \le \varepsilon, \quad \text{for } x \in [0, 1].$$

Now we consider the cases for $2 \le m \le k$. We assume for $F_{m-1}$, there exists a multilayer neural network $\tilde{F}_{m-1}$ with not more than $T_1(m-1) \log_3 \frac{3^m}{\varepsilon}$ layers, $T_2(m-1) \log_3 \frac{3^m}{\varepsilon}$ binary step units and $T_3(m-1) \left( \log_3 \frac{3^m}{\varepsilon} \right)^2$ rectifier linear units such that

$$|F_{m-1}(x) - \tilde{F}_{m-1}(x)| \le \frac{\varepsilon}{3}, \quad \text{for } x \in [0, 1].$$

Further we assume the derivative of $F_{m-1}$ has an upper bound $\left\|F'_{m-1}\right\| \leq 1$. Then for $F_m$, since $F_m(x)$ can be rewritten as

$$F_m(x) = F_{m-1}(h_m(x)),$$

and there exists a multilayer neural network $\tilde{h}_m$ with at most $T_1(1) \log_3 \frac{3}{\varepsilon}$ layers, $T_2(1) \log_3 \frac{3}{\varepsilon}$ binary step units and $T_3(1) \left(\log_3 \frac{3}{\varepsilon}\right)^2$ rectifier linear units such that

$$|h_m(x) - \tilde{h}_m(x)| \leq \frac{\varepsilon}{3}, \quad \text{for } x \in [0, 1],$$

and $\left\|\tilde{h}_m\right\| \leq (1 + \varepsilon/3)$. Then for cascaded multilayer neural network $\tilde{F}_m = \tilde{F}_{m-1} \circ \left(\frac{1}{1+\varepsilon/3}\tilde{h}_m\right)$, we have

$$\| F_m - \tilde{F}_m \| = \left\|F_{m-1}(h_m) - \tilde{F}_{m-1}\left(\frac{\tilde{h}_m}{1+\varepsilon/3}\right)\right\|$$

$$\leq \left\|F_{m-1}(h_m) - F_{m-1}\left(\frac{\tilde{h}_m}{1+\varepsilon/3}\right)\right\| + \left\|F_{m-1}\left(\frac{\tilde{h}_m}{1+\varepsilon/3}\right) - \tilde{F}_{m-1}\left(\frac{\tilde{h}_m}{1+\varepsilon/3}\right)\right\|$$

$$\leq \left\|F'_{m-1}\right\| \cdot \left\|h_m - \frac{\tilde{h}_m}{1+\varepsilon/3}\right\| + \frac{\varepsilon}{3}$$

$$\leq \left\|F'_{m-1}\right\| \cdot \left\|h_m - \tilde{h}_m\right\| + \left\|F'_{m-1}\right\| \cdot \left\|\frac{\varepsilon/3}{1+\varepsilon/3}\tilde{h}_m\right\| + \frac{\varepsilon}{3}$$

$$\leq \frac{\varepsilon}{3} + \frac{\varepsilon}{3} + \frac{\varepsilon}{3} = \varepsilon$$

In addition, the derivative of $F_m$ can be upper bounded by

$$\left\|F'_m\right\| \leq \left\|F'_{m-1}\right\| \cdot \left\|h'_m\right\| = 1.$$

Since the multilayer neural network $\tilde{F}_m$ is constructed by cascading multilayer neural networks $\tilde{F}_{m-1}$ and $\tilde{h}_m$, then the iterations for $T_1$, $T_2$ and $T_3$ are

$$T_1(m) \log_3 \frac{3^m}{\varepsilon} = T_1(m-1) \log_3 \frac{3^m}{\varepsilon} + T_1(1) \log_3 \frac{3}{\varepsilon}, \tag{4}$$

$$T_2(m) \log_3 \frac{3^m}{\varepsilon} = T_2(m-1) \log_3 \frac{3^m}{\varepsilon} + T_2(1) \log_3 \frac{3}{\varepsilon}, \tag{5}$$

$$T_3(m) \left(\log_3 \frac{3^m}{\varepsilon}\right)^2 = T_3(m-1) \left(\log_3 \frac{3^m}{\varepsilon}\right)^2 + T_3(1) \left(\log_3 \frac{3}{\varepsilon}\right)^2. \tag{6}$$

From iterations (4) and (5), we could have for $2 \leq m \leq k$,

$$T_1(m) = T_1(m-1) + T_1(1)\frac{1 + \log_3(1/\varepsilon)}{m + \log_3(1/\varepsilon)} \leq T_1(m-1) + T_1(1)\frac{1 + \log_3(1/\varepsilon)}{m}$$

$$T_2(m) = T_2(m-1) + T_2(1)\frac{1 + \log_3(1/\varepsilon)}{m + \log_3(1/\varepsilon)} \leq T_2(m-1) + T_2(1)\frac{1 + \log_3(1/\varepsilon)}{m}$$

and thus

$$T_1(k) = \mathcal{O}\left(\log k \log \frac{1}{\varepsilon}\right), \quad T_2(k) = \mathcal{O}\left(\log k \log \frac{1}{\varepsilon}\right).$$

From the iteration (6), we have for $2 \leq m \leq k$,

$$T_3(m) = T_3(m-1) + T_3(1) \left(\frac{1 + \log_3(1/\varepsilon)}{m + \log_3(1/\varepsilon)}\right)^2 \leq T_3(m-1) + \frac{(1 + \log_3(1/\varepsilon))^3}{m^2},$$

and thus

$$T_3(k) = \mathcal{O}\left(\left(\log \frac{1}{\varepsilon}\right)^2\right).$$

Therefore, to approximate $f = F_k$, we need at most $\mathcal{O}\left(k \log k \log \frac{1}{\varepsilon} + \log k \left(\log \frac{1}{\varepsilon}\right)^2\right)$ layers, $\mathcal{O}\left(k \log k \log \frac{1}{\varepsilon} + \log k \left(\log \frac{1}{\varepsilon}\right)^2\right)$ binary step units and $\mathcal{O}\left(k^2 \left(\log \frac{1}{\varepsilon}\right)^2 + \left(\log \frac{1}{\varepsilon}\right)^4\right)$ rectifier linear units.

$\square$

## APPENDIX D   PROOF OF THEOREM 8

*Proof.* The proof is composed of two parts. As before, we first use the deep structure shown in Figure 1 to find the binary expansion of $\boldsymbol{x}$ and next use a multilayer neural network to approximate the polynomial.

Let $\boldsymbol{x} = (x^{(1)}, ..., x^{(d)})$ and $\boldsymbol{w}_i = (w_{i1}, ..., w_{id})$. We could now use the deep structure shown in Figure 1 to find the binary expansion for each $x^{(k)}$, $k \in [d]$. Let $\tilde{x}^{(k)} = \sum_{r=0}^{n} \frac{x_r^{(k)}}{2^r}$ denote the binary expansion of $x^{(k)}$, where $x_r^{(k)}$ is the $r$th bit in the binary expansion of $x^{(k)}$. Obviously, to decode all the $n$-bit binary expansions of all $x^{(k)}$, $k \in [d]$, we need a multilayer neural network with $n$ layers and $dn$ binary units in total. Besides, we let $\tilde{\boldsymbol{x}} = (\tilde{x}^{(1)}, ..., \tilde{x}^{(d)})$. Now we define

$$\tilde{f}(\boldsymbol{x}) = f(\tilde{\boldsymbol{x}}) = \prod_{i=1}^{p} \left( \sum_{k=1}^{d} w_{ik}\tilde{x}^{(k)} \right).$$

We further define

$$g_l(\tilde{\boldsymbol{x}}) = \prod_{i=1}^{l} \left( \sum_{k=1}^{d} w_{ik}\tilde{x}^{(k)} \right).$$

Since for $l = 1, ..., p - 1$,

$$g_l(\tilde{\boldsymbol{x}}) = \prod_{i=1}^{l} \left( \sum_{k=1}^{d} w_{ik}\tilde{x}^{(k)} \right) \leq \prod_{i=1}^{l} \|\boldsymbol{w}_i\|_1 = 1,$$

then we can rewrite $g_{l+1}(\tilde{\boldsymbol{x}})$, $l = 1, ..., p - 1$ into

$$g_{l+1}(\tilde{\boldsymbol{x}}) = \prod_{i=1}^{l+1} \left( \sum_{k=1}^{d} w_{ik}\tilde{x}^{(k)} \right) = \sum_{k=1}^{d} \left[ w_{(l+1)k}\tilde{x}^{(k)} \cdot g_l(\tilde{\boldsymbol{x}}) \right] = \sum_{k=1}^{d} \left\{ w_{(l+1)k} \sum_{r=0}^{n} \left[ x_r^{(k)} \cdot \frac{g_l(\tilde{\boldsymbol{x}})}{2^r} \right] \right\}$$
$$= \sum_{k=1}^{d} \left\{ w_{(l+1)k} \sum_{r=0}^{n} \max \left[ 2(x_r^{(k)} - 1) + \frac{g_l(\tilde{\boldsymbol{x}})}{2^r}, 0 \right] \right\} \tag{7}$$

Obviously, equation (7) defines a relationship between the outputs of neighbor layers and thus can be used to implement the multilayer neural network. In this implementation, we need $dn$ rectifier linear units in each layer and thus $dnp$ rectifier linear units. Therefore, to implement function $\tilde{f}(\boldsymbol{x})$, we need $p + n$ layers, $dn$ binary step units and $dnp$ rectifier linear units in total.

In the rest of proof, we consider the approximation error. Since for $k = 1, ..., d$ and $\forall \boldsymbol{x} \in [0, 1]^d$,

$$\left| \frac{\partial f(\boldsymbol{x})}{\partial x^{(k)}} \right| = \left| \sum_{j=1}^{p} \left[ w_{jk} \cdot \prod_{i=1, i \neq j}^{p} (\boldsymbol{w}_i^T \boldsymbol{x}) \right] \right| \leq \sum_{j=1}^{p} |w_{jk}| \leq p,$$

then

$$|f(\boldsymbol{x}) - \tilde{f}(\boldsymbol{x})| = |f(\boldsymbol{x}) - f(\tilde{\boldsymbol{x}})| \leq \|\nabla f\|_2 \cdot \|\boldsymbol{x} - \tilde{\boldsymbol{x}}\|_2 \leq \frac{pd}{2^n}.$$

By choosing $n = \left\lceil \log_2 \frac{pd}{\varepsilon} \right\rceil$, we have

$$|f(\boldsymbol{x}) - f(\tilde{\boldsymbol{x}})| \leq \varepsilon.$$

Since we use $nd$ binary step units to convert the input to binary form and $dnp$ neurons in function approximation, we thus use $\mathcal{O}\left( d \log \frac{pd}{\varepsilon} \right)$ binary step units and $\mathcal{O}\left( pd \log \frac{pd}{\varepsilon} \right)$ rectifier linear units in total. In addition, since we have used $n$ layers to convert the input to binary form and $p$ layers in the function approximation section of the network, the whole deep structure has $\mathcal{O}\left( p + \log \frac{pd}{\varepsilon} \right)$ layers. $\square$

## APPENDIX E  PROOF OF THEOREM 9

*Proof.* For each multinomial function $g$ with multi-index $\boldsymbol{\alpha}$, $g_{\boldsymbol{\alpha}}(\boldsymbol{x}) = \boldsymbol{x}^{\boldsymbol{\alpha}}$, it follows from Theorem 4 that there exists a deep neural network $\tilde{g}_{\boldsymbol{\alpha}}$ of size $\mathcal{O}\left(|\boldsymbol{\alpha}| \log \frac{|\boldsymbol{\alpha}| d}{\varepsilon}\right)$ and depth $\mathcal{O}\left(|\boldsymbol{\alpha}| + \log \frac{|\boldsymbol{\alpha}| d}{\varepsilon}\right)$ such that

$$|g_{\boldsymbol{\alpha}}(\boldsymbol{x}) - \tilde{g}_{\boldsymbol{\alpha}}(\boldsymbol{x})| \leq \varepsilon.$$

Let the deep neural network be

$$\tilde{f}(\boldsymbol{x}) = \sum_{\boldsymbol{\alpha}:|\boldsymbol{\alpha}|\leq p} C_{\boldsymbol{\alpha}} \tilde{g}_{\boldsymbol{\alpha}}(\boldsymbol{x}),$$

and thus

$$|f(\boldsymbol{x}) - \tilde{f}(\boldsymbol{x})| \leq \sum_{\boldsymbol{\alpha}:|\boldsymbol{\alpha}|\leq p} |C_{\boldsymbol{\alpha}}| \cdot |g_{\boldsymbol{\alpha}}(\boldsymbol{x}) - \tilde{g}_{\boldsymbol{\alpha}}(\boldsymbol{x})| = \varepsilon.$$

Since the total number of multinomial is upper bounded by

$$p \binom{p+d-1}{d-1},$$

the size of deep neural network is thus upper bounded by

$$p^2 \binom{p+d-1}{d-1} \log \frac{pd}{\varepsilon}. \tag{8}$$

If the dimension of the input $d$ is fixed, then (8) is has the order of

$$p^2 \binom{p+d-1}{d-1} \log \frac{pd}{\varepsilon} = \mathcal{O}\left((ep)^{d+1} \log \frac{pd}{\varepsilon}\right), \quad p \to \infty$$

while if the degree $p$ is fixed, then (8) is has the order of

$$p^2 \binom{p+d-1}{d-1} \log \frac{pd}{\varepsilon} = \mathcal{O}\left(p^2 (ed)^p \log \frac{pd}{\varepsilon}\right), \quad d \to \infty.$$

$\square$

## APPENDIX F  PROOF OF THEOREM 11

*Proof.* We first prove the univariate case $d = 1$. The proof is composed of two parts. We say the function $g(x)$ has a *break point* at $x = z$ if $g$ is discontinuous at $z$ or its derivative $g'$ is discontinuous at $z$. We first present the lower bound on the number of break points $M(\varepsilon)$ that the multilayer neural network $\tilde{f}$ should have for $\varepsilon$-approximation of function $f$ with error $\varepsilon$. We next relate the number of break points $M(\varepsilon)$ to the network depth $L$ and the size $N$.

Now we calculate the lower bound on $M(\varepsilon)$. We first define 4 points $x_0$, $x_1 = x_0 + 2\sqrt{\rho\varepsilon/\mu}$, $x_2 = x_1 + 2\sqrt{\rho\varepsilon/\mu}$ and $x_3 = x_2 + 2\sqrt{\rho\varepsilon/\mu}$, $\forall \rho > 1$. We assume

$$0 \leq x_0 < x_1 < x_2 < x_3 \leq 1.$$

We now prove that if multilayer neural network $\tilde{f}$ has no break point in $[x_1, x_2]$, then $\tilde{f}$ should have a break point in $[x_0, x_1]$ and a break point in $[x_2, x_3]$. We prove this by contradiction. We assume the neural network $\tilde{f}$ has no break points in the interval $[x_0, x_3]$. Since $\tilde{f}$ is constructed by rectifier linear units and binary step units and has no break points in the interval $[x_0, x_3]$, then $\tilde{f}$ should be a linear function in the interval $[x_0, x_3]$, *i.e.*, $\tilde{f}(x) = ax + b$, $x \in [x_0, x_3]$ for some $a$ and $b$. By assumption, since $\tilde{f}$ approximates $f$ with error at most $\varepsilon$ everywhere in $[0, 1]$, then

$$|f(x_1) - ax_1 - b| \leq \varepsilon \quad \text{and} \quad |f(x_2) - ax_2 - b| \leq \varepsilon.$$

Then we have

$$\frac{f(x_2) - f(x_1) - 2\varepsilon}{x_2 - x_1} \leq a \leq \frac{f(x_2) - f(x_1) + 2\varepsilon}{x_2 - x_1}.$$

By strong convexity of $f$,

$$\frac{f(x_2) - f(x_1)}{x_2 - x_1} + \frac{\mu}{2}(x_2 - x_1) \leq f'(x_2).$$

Besides, since $\rho > 1$ and

$$\frac{\mu}{2}(x_2 - x_1) = \sqrt{\rho\mu\varepsilon} = \frac{2\rho\varepsilon}{x_2 - x_1} > \frac{2\varepsilon}{x_2 - x_1},$$

then

$$a \leq f'(x_2). \tag{9}$$

Similarly, we can obtain $a \geq f'(x_1)$. By our assumption that $\tilde{f} = ax + b$, $x \in [x_0, x_3]$, then

$$
\begin{aligned}
f(x_3) - \tilde{f}(x_3) &= f(x_3) - ax_3 - b \\
&= f(x_3) - f(x_2) - a(x_3 - x_2) + f(x_2) - ax_2 - b \\
&\geq f'(x_2)(x_3 - x_2) + \frac{\mu}{2}(x_3 - x_2)^2 - a(x_3 - x_2) - \varepsilon \\
&= (f'(x_2) - a)(x_3 - x_2) + \frac{\mu}{2}\left(2\sqrt{\rho\varepsilon/\mu}\right)^2 - \varepsilon \\
&\geq (2\rho - 1)\varepsilon > \varepsilon
\end{aligned}
$$

The first inequality follows from strong convexity of $f$ and $f(x_2) - ax_2 - b \geq \varepsilon$. The second inequality follows from the inequality (9). Therefore, this leads to the contradiction. Thus there exists a break point in the interval $[x_2, x_3]$. Similarly, we could prove there exists a break point in the interval $[x_0, x_1]$. These indicate that to achieve $\varepsilon$-approximation in $[0, 1]$, the multilayer neural network $\tilde{f}$ should have at least $\left\lceil \frac{1}{4}\sqrt{\frac{\mu}{\rho\varepsilon}} \right\rceil$ break points in $[0, 1]$. Therefore,

$$M(\varepsilon) \geq \left\lceil \frac{1}{4}\sqrt{\frac{\mu}{\rho\varepsilon}} \right\rceil, \quad \forall \rho > 1.$$

Further, Telgarsky (2016) has shown that the maximum number of break points that a multilayer neural network of depth $L$ and size $N$ could have is $(N/L)^L$. Thus, $L$ and $N$ should satisfy

$$(N/L)^L > \left\lceil \frac{1}{4}\sqrt{\frac{\mu}{\rho\varepsilon}} \right\rceil, \quad \forall \rho > 1.$$

Therefore, we have

$$N \geq L\left(\frac{\mu}{16\varepsilon}\right)^{\frac{1}{2L}}.$$

Besides, let $m = N/L$. Since each layer in network should have at least 2 neurons, *i.e.*, $m \geq 2$, then

$$N \geq \frac{m}{2\log_2 m}\log_2\left(\frac{\mu}{16\varepsilon}\right) \geq \log_2\left(\frac{\mu}{16\varepsilon}\right).$$

Now we consider the multivariate case $d > 1$. Assume input vector to be $\boldsymbol{x} = (x^1, ..., x^{(d)})$. We now fix $x^{(2)}, ..., x^{(d)}$ and define two univariate functions

$$g(y) = f(y, x^{(2)}, ..., x^{(d)}), \text{ and } \tilde{g}(y) = \tilde{f}(y, x^{(2)}, ..., x^{(d)}).$$

By assumption, $g(y)$ is a strongly convex function with parameter $\mu$ and for all $y \in [0, 1]$, $|g(y) - \tilde{g}(y)| \leq \varepsilon$. Therefore, by results in the univariate case, we should have

$$N \geq L\left(\frac{\mu}{16\varepsilon}\right)^{\frac{1}{2L}} \quad \text{and} \quad N \geq \log_2\left(\frac{\mu}{16\varepsilon}\right). \tag{10}$$

Now we have proved the theorem.

**Remark:** We make the following remarks about the lower bound in the theorem.

(1) if the depth $L$ is fixed, as in shallow networks, the number of neurons required is $\Omega\left((1/\varepsilon)^{\frac{1}{2L}}\right)$.

(2) if we are allowed to choose $L$ optimally to minimize the lower bound, we will choose $L = \frac{1}{2}\log(\frac{\mu}{16\varepsilon})$ and thus the lower bound will become $\Omega(\log\frac{1}{\varepsilon})$, closed to the $\mathcal{O}(\log^2\frac{1}{\varepsilon})$ upper bound shown in Theorem 4.

$\square$

## APPENDIX G   PROOF OF COROLLARY 12

*Proof.* From Theorem 4, it follows that there exists a deep neural network $\tilde{f}_d$ of depth $L_d = \Theta\left(\log\frac{1}{\varepsilon}\right)$ and size

$$N_d \leq c\left(\log\frac{1}{\varepsilon}\right)^2 \tag{11}$$

for some constant $c > 0$ such that $\|\tilde{f}_d - f\| \leq \varepsilon$.

From the equation (10) in the proof of Theorem 11, it follows that for all shallow neural networks $\tilde{f}_s$ of depth $L_s$ and $\left\|\tilde{f}_s - f\right\| \leq \varepsilon$, their sizes should satisfy

$$N_s \geq L_s\left(\frac{\mu}{16\varepsilon}\right)^{\frac{1}{2L_s}},$$

which is equivalent to

$$\log N_s \geq \log L_s + \frac{1}{2L_s}\log\left(\frac{\mu}{16\varepsilon}\right). \tag{12}$$

Substituting for $\log\left(\frac{1}{\varepsilon}\right)$ from (12) to (11), we have

$$N_d = \mathcal{O}(L_s^2 \log^2 N_s).$$

By definition, a shallow neural network has a small number of layers, i.e., $L_s$. Thus, the size of the deep neural network is $\mathcal{O}(\log^2 N_s)$. This means $N_d \ll N_s$.   $\square$

## APPENDIX H   PROOF OF COROLLARY 13

**Corollary 13** (Gaussian function). *For Gaussian function* $f(\boldsymbol{x}) = f(x^{(1)}, ..., x^{(d)}) = e^{-\sum_{i=1}^{d}(x^{(i)})^2/2}$, $\boldsymbol{x} \in [0,1]^d$, *there exists a deep neural network* $\tilde{f}(\boldsymbol{x})$ *with* $\mathcal{O}\left(\log\frac{d}{\varepsilon}\right)$ *layers,* $\mathcal{O}\left(d\log\frac{d}{\varepsilon}\right)$ *binary step units and* $\mathcal{O}\left(d\log\frac{d}{\varepsilon} + \left(\log\frac{1}{\varepsilon}\right)^2\right)$ *rectifier linear units such that* $|\tilde{f}(\boldsymbol{x}) - f(\boldsymbol{x})| \leq \varepsilon$ *for* $\forall \boldsymbol{x} \in [0,1]^d$.

*Proof.* It follows from the Theorem 4 that there exists $d$ multilayer neural networks $\tilde{g}_1(x^{(1)}), ..., \tilde{g}_d(x^{(d)})$ with $\mathcal{O}\left(\log\frac{d}{\varepsilon}\right)$ layers and $\mathcal{O}\left(d\log\frac{d}{\varepsilon}\right)$ binary step units and $\mathcal{O}\left(d\log\frac{d}{\varepsilon}\right)$ rectifier linear units in total such that

$$\left|\frac{x^{(1)^2} + ... + x^{(d)^2}}{2} - \frac{\tilde{g}_1(x^{(1)}) + ... + \tilde{g}_d(x^{(d)})}{2}\right| \leq \frac{\varepsilon}{2}. \tag{13}$$

Besides, from Theorem 4, it follows that there exists a deep neural network $\hat{f}$ with $\mathcal{O}\left(\log\frac{1}{\varepsilon}\right)$ layers $\mathcal{O}\left(\log\frac{1}{\varepsilon}\right)$ binary step units and $\mathcal{O}\left(\left(\log\frac{1}{\varepsilon}\right)^2\right)$ such that

$$|e^{-dx} - \hat{f}(x)| \leq \frac{\varepsilon}{2}, \quad \forall x \in [0,1].$$

Let $x = (\tilde{g}_1(x^{(1)}) + ... + \tilde{g}_d(x^{(d)}))/2d$, then we have

$$\left|e^{-(\sum_{i=1}^{d}\tilde{g}_i(x^{(i)}))/2} - \hat{f}\left(\frac{\sum_{i=1}^{d}\tilde{g}_i(x^{(i)})}{2}\right)\right| \leq \frac{\varepsilon}{2}. \tag{14}$$

Let the deep neural network

$$\tilde{f}(\boldsymbol{x}) = \hat{f}\left(\frac{\tilde{g}_1(x^{(1)}) + ... + \tilde{g}_d(x^{(d)})}{2}\right).$$

By inequalities (13) and (14), the the approximation error is upper bounded by

$$|f(\boldsymbol{x}) - \tilde{f}(\boldsymbol{x})| = \left| e^{-(\sum_{i=1}^{d} x^{(i)})/2} - \hat{f}\left( \frac{\sum_{i=1}^{d} \tilde{g}_i(x^{(i)})}{2} \right) \right|$$

$$\leq \left| e^{-(\sum_{i=1}^{d} x^{(i)})/2} - e^{-(\sum_{i=1}^{d} \tilde{g}_i(x^{(i)}))/2} \right| + \left| e^{-(\sum_{i=1}^{d} \tilde{g}_i(x^{(i)}))/2} - \hat{f}\left( \frac{\sum_{i=1}^{d} \tilde{g}_i(x^{(i)})}{2} \right) \right|$$

$$\leq \frac{\varepsilon}{2} + \frac{\varepsilon}{2} = \varepsilon.$$

Now the deep neural network has $\mathcal{O}\left(\log \frac{d}{\varepsilon}\right)$ layers, $\mathcal{O}\left(d \log \frac{d}{\varepsilon}\right)$ binary step units and $\mathcal{O}\left(d \log \frac{d}{\varepsilon} + \left(\log \frac{1}{\varepsilon}\right)^2\right)$ rectifier linear units.

$\square$

## APPENDIX I   PROOF OF COROLLARY 14

**Corollary 14** (Ridge function). *If $f(\boldsymbol{x}) = g(\boldsymbol{a}^T \boldsymbol{x})$ for some direction $\boldsymbol{a} \in \mathbb{R}^d$ with $\|\boldsymbol{a}\|_1 = 1$, $\boldsymbol{a} \succeq \boldsymbol{0}$, $\boldsymbol{x} \in [0,1]^d$ and some univariate function $g$ satisfying conditions in Theorem 4, then there exists a multilayer neural network $\tilde{f}$ with $\mathcal{O}\left(\log \frac{1}{\varepsilon}\right)$ layers, $\mathcal{O}\left(\log \frac{1}{\varepsilon}\right)$ binary step units and $\mathcal{O}\left(\left(\log \frac{1}{\varepsilon}\right)^2\right)$ rectifier linear units such that $|f(\boldsymbol{x}) - \tilde{f}(\boldsymbol{x})| \leq \varepsilon$ for $\forall \boldsymbol{x} \in [0,1]^d$.*

*Proof.* Let $t = \boldsymbol{a}^T \boldsymbol{x}$. Since $\|\boldsymbol{a}\|_1 = 1$, $\boldsymbol{a} \succeq \boldsymbol{0}$ and $\boldsymbol{x} \in [0,1]^d$, then $0 \leq t \leq 1$. Then from Theorem 4, it follows that then there exists a multilayer neural network $\tilde{g}$ with $\mathcal{O}\left(\log \frac{1}{\varepsilon}\right)$ layers, $\mathcal{O}\left(\log \frac{1}{\varepsilon}\right)$ binary step units and $\mathcal{O}\left(\left(\log \frac{1}{\varepsilon}\right)^2\right)$ rectifier linear units such that

$$|g(t) - \tilde{g}(t)| \leq \varepsilon, \quad \forall t \in [0,1].$$

If we define the deep network $\tilde{f}$ as

$$\tilde{f}(\boldsymbol{x}) = \tilde{g}(t),$$

then the approximation error of $\tilde{f}$ is

$$|f(\boldsymbol{x}) - \tilde{f}(\boldsymbol{x})| = |g(t) - \tilde{g}(t)| \leq \varepsilon.$$

Now we have proved the corollary.

$\square$

