# Peer review of "Why Deep Neural Networks for Function Approximation?"

_ICLR 2017 — accepted_

[Official Review · AnonReviewer2 · rating 7 · confidence 3 · 14 Dec 2016]
**No Title**

This paper shows:

  1. Easy, constructive proofs to derive e-error upper-bounds on neural networks with O(log 1/e) layers and O(log 1/e) ReLU units.
  2. Extensions of the previous results to more general function classes, such as smooth or vector-valued functions.
  3. Lower bounds on the neural network size, as a function of its number of layers. The lower bound reveals the need of exponentially many more units to approximate functions using shallow architectures.

The paper is well written and easy to follow. The technical content, including the proofs in the Appendix, look correct. Although the proof techniques are simple (and are sometimes modifications of arguments by Gil, Telgarsky, or Dasgupta), they are brought together in a coherent manner to produce sharp results. Therefore, I am leaning toward acceptance.

[Official Review · AnonReviewer1 · rating 7 · confidence 4 · 16 Dec 2016]
**No Title**

The main contribution of this paper is a construction to eps-approximate a piecewise smooth function with a multilayer neural network that uses O(log(1/eps)) layers and O(poly log(1/eps)) hidden units where the activation functions can be either ReLU or binary step or any combination of them. The paper is well written and clear. The arguments and proofs are easy to follow. I only have two questions:

1- It would be great to have similar results without binary step units. To what extent do you find the binary step unit central to the proof?

2- Is there an example of piecewise smooth function that requires at least poly(1/eps) hidden units with a shallow network?

[Official Review · AnonReviewer3 · rating 7 · confidence 4 · 20 Dec 2016]
**review of ``WHY DEEP NEURAL NETWORKS FOR FUNCTION APPROXIMATION?''**

SUMMARY 
This paper contributes to the description and comparison of the representational power of deep vs shallow neural networks with ReLU and threshold units. The main contribution of the paper is to show that approximating a strongly convex differentiable function is possible with much less units when using a network with one more hidden layer. 

PROS 
The paper presents an interesting combination of tools and arrives at a nice result on the exponential superiority of depth. 

CONS
The main result appears to address only strongly convex univariate functions. 

SPECIFIC COMMENTS 

- Thanks for the comments on L. Still it would be a good idea to clarify this point as far as possible in the main part. Also, I would suggest to advertise the main result more prominently. 
I still have not read the revision and maybe you have already addressed some of these points there. 

- The problem statement is close to that from [Montufar, Pascanu, Cho, Bengio NIPS 2014], which specifically arrives at exponential gaps between deep and shallow ReLU networks, albeit from a different angle. I would suggest to include that paper it in the overview. 

- In Lemma 3, there is an i that should be x

- In Theorem 4, ``\tilde f'' is missing the (x). 

- Theorem 11, the lower bound always increases with L ? 

- In Theorem 11, \bf x\in [0,1]^d?

[Final Decision · Program Chairs · 06 Feb 2017]
**ICLR committee final decision**

The paper makes a solid technical contribution in proving that the deep networks are exponentially more efficient in function approximation compared to the shallow networks. They take the case of piecewise smooth networks, which is practically motivated (e.g. images have edges with smooth regions), and analyze the size of both the deep and shallow networks required to approximate it to the same degree.
 
 The reviewers recommend acceptance of the paper and I am happy to go with their recommendation.